# Analysis of Composite Structures in Curing Process for Shape Deformations and Shear Stress: Basis for Advanced Optimization

**Niraj Kumbhare** [1], **Reza Moheimani** [1,2] and **Hamid Dalir** [1,*]

1   Advanced Composite Structures Engineering Laboratory, Department of Mechanical and Energy Engineering, Purdue School of Engineering and Technology, Indianapolis, IN 46202, USA; nirkumbh@iupui.edu (N.K.); rezam@purdue.edu (R.M.)
2   School of Mechanical Engineering, Purdue University, West Lafayette, IN 47907, USA
*   Correspondence: hdalir@purdue.edu

**Abstract:** Identifying residual stresses and the distortions in composite structures during the curing process plays a vital role in coming up with necessary compensations in the dimensions of mold or prototypes and having precise and optimized parts for the manufacturing and assembly of composite structures. This paper presents an investigation into process-induced shape deformations in composite parts and structures, as well as a comparison of the analysis results to finalize design parameters with a minimum of deformation. A Latin hypercube sampling (LHS) method was used to generate the required random points of the input variables. These variables were then executed with the Ansys Composite Cure Simulation (ACCS) tool, which is an advanced tool used to find stress and distortion values using a three-step analysis, including Ansys Composite PrepPost, transient thermal analysis, and static structural analysis. The deformation results were further utilized to find an optimum design to manufacture a complex composite structure with the compensated dimensions. The simulation results of the ACCS tool are expected to be used by common optimization techniques to finalize a prototype design so that it can reduce common manufacturing errors like warpage, spring-in, and distortion.

**Keywords:** advanced composite cure simulation; carbon fiber-reinforced composites; optimization study; process-induced shape deformation; design of experiments; Latin hypercube sampling

## 1. Introduction

Composite structures have a vast range of applications in the aerospace, automobile, and energy industries. As time has passed, the manufacturing of composites has grown exponentially. The necessity of lightweight design and higher performance in the aerospace and motorsports industries has increased the demand for carbon fiber-reinforced polymer composite materials. Residual stresses and shape deformations are the main obstacles for high performance in a composite structure. The use of deformed components in assembly can cause higher internal stresses, which then hamper a final product's performance [1–6].

The thermomechanical analysis of composites is a hot topic for all researchers in the composite world. An ample amount of data have recently been collected from innovative research on composite structures' behavior. However, there is still a lack of standards to measure deformation values and residual stresses during the curing process of composite manufacturing [7,8]. In the automobile industry, tool designers decide the parameters based on trial results and their experienced guesses to estimate distortion and warpage. The most frequent problem found in the approximation method is that the results are not precise for intricate designs [9,10]. Finding optimum design parameters for tool development becomes a tedious task because of these deformations and residual stresses. The need for a reliable approach to predict these shape distortions has driven researchers to develop new methods to find deformation values [7,11,12].

A variety of research has been done on the process-induced shape deformations and stresses of nonplanar parts. However, thin planar composite plates show complex deformations that cannot be verified with old methods. Ersoy et al. [11] implemented a two-step finite element analysis method to calculate curved parts' deformations. Wisnom and M.R. [12] found that the spring-in angle of curved composites is proportional to the laminate's through-thickness. Zhang, G. and J. Wang [13] represented the results of an investigation on the process-induced stress and deformation of variable-stiffness composite cylinders. Mezeix et al. [14] presented a method to predict composite flight structures' deformation using ABAQUS software

Ansys composite cure simulation is a novel method developed using the Ansys tool, which was used as a part of the work presented in this paper. Curing process analysis is the building block for obtaining accurate results for predicting the final shape of a composite part [15]. G. Fernlund [16] explained the significance of the cure cycle in the dimensional fidelity of autoclave-processed composite parts. The final laminate quality is dependent on the heating rate, initial cure temperature, and dwelling time [17,18]. Temperature has been proven to leave a significant effect on the body of an alloy while using finite element analysis (FEA) [19]. The coefficient of thermal expansion (CTE) is different in different directions for composite materials. Previous experimental studies have shown that the CTE of resin-dominated directions is much higher than that of fiber-dominated directions [20,21]. The stresses get built up on the fiber-matrix, lamina–laminate, and structural levels. When a composite part is cooled, the generated residual stresses are compensated for by the tool dimensions when they are in contact with the tool. The piece gets distorted to its equilibrium state to balance the internal residual stresses when the support gets removed [22]. In a literature survey, it was found that some of the factors responsible for creating deformations are the layup sequence, resin cure shrinkage, tool–part interaction, part angles, curing time and cycle, ply thickness, and layup angles.

The next step is to analyze simulation results to compare and find optimized parameters with an objective of minimum deformation. Efficient design is obtained by sizing a composite part's geometry, altering the manufacturing process, and tailoring design variables that control mechanical properties such as fiber orientation, the number of plies, and the stacking sequence. The manufacturability of a virtually optimized composite structure is a crucial precondition for the usability of the best results. In the current work, the consideration of manufacturing constraints ensured that all compared solutions are producible. This paper shows a study of the fundamentals of the design optimization of composite parts by considering manufacturing limitations. The optimization of composite parts that combine finite element (FE) models takes a has-run time for each FE simulation [23]. Generally, composite design optimization is a non-convex, multimodal optimization problem that involves continuous and discrete variables. In such cases, population-based algorithms like genetic algorithms (GAs) are preferred because they use several design responses in each iteration to find the optimal solution instead of gradient information. In the current research work, the simulation results were analyzed to prepare the base of an optimization algorithm [24–26]. Amir Ehsani [27] demonstrated the GA technique used to optimize composite angle grid plates.

This paper presents a reliable engineering approach to find process-induced shape deformations using the Ansys Composite Cure Simulation (ACCS) tool, utilizing the results of its analysis to find an optimal design with the least deformation using the fundamentals of design optimization. A novel FEA method is proposed to address the challenge of predicting residual stresses, distorted values, and the importance of the study of composites' thermal behavior. The global optimization gave optimal model results, which helped to finalize the design.

## 2. Materials and Methods

An optimal design can be found by combining ACCS simulation results with a standard optimization process. ACCS has been used to analyze deformations like spring-in,

warpage, and residual stresses in composite parts. Deformation results vary with different inputs, including the layup sequence, angles, support constraint, and cure cycle. These variations were studied to find the best design to manufacture a complex composite structure. Figure 1 illustrates the full methodology starting from the selection of design samples to the final optimized design selection. The process started with the design optimization method, which was further divided into sub-processes, such as the objective function selection, constraints, and design of experiments (DOEs). Executing FEA was the second important step, which included the simulation process to find deformation results. These results were further analyzed and compared to find the optimized design. The methods explained in this section include all the details about the simulation and optimization process.

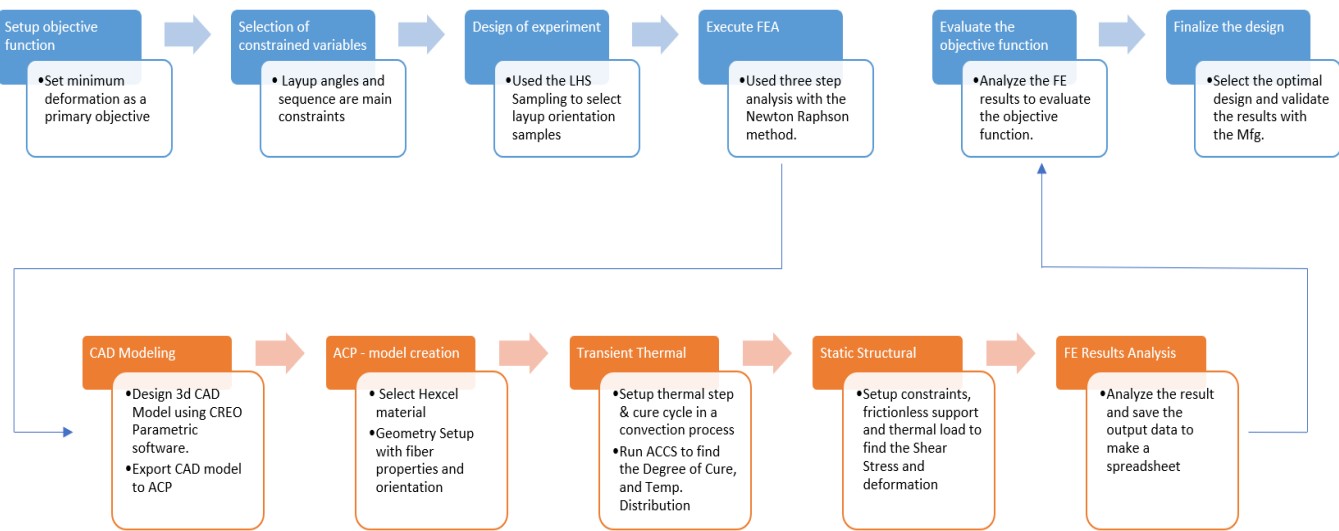

**Figure 1.** Process flow chart to find the final design. ACP: ANSYS Composite PrepPost; LHS: Latin hypercube sampling; FE: finite element; ACCS: Ansys Composite Cure Simulation.

### 2.1. Materials

The automotive and aerospace industry uses various composite materials, including natural composites, carbon–carbon composites, metal matrix composites (MMCs), and polymer matrix composites (PMCs). They provide benefits such as weight reduction, durability, high strength, and energy absorption. In this project, a Hexcel AS4-8552 prepreg—a unidirectional prepreg with a high-performance polymer matrix—was used for the simulation process. Hexcel material can make it possible to achieve weight reductions by maintaining a component's high structural performance. Using Hexcel AS4-8552 was beneficial because of its properties like high impact resistance, reasonable translation of fiber properties, and high strength. The mechanical properties of the Hexcel AS4 material provided by the manufacturer are mentioned in Table 1 [15]. The *X* and *Y* directions indicate the fiber directions as parallel (0°) and transverse (90°), respectively, to the matrix. Ansys has the feature of defining customized material properties in the engineering data section. These properties were applied to the L plate of dimensions 50 × 86 × 1.6 mm with a flange height of 50 mm.

**Table 1.** Material properties of Hexcel AS4-8552.

| Material Property | Value |
|:---:|:---:|
| Density | 1580 kg/m$^3$ |
| **Coefficient of Thermal Expansion** | |
| i.  X-Direction | $1 \times 10^{-20}/°C$ |
| ii.  Y/Z-Direction | $3.261 \times 10^{-5}/°C$ |
| **Young's Modulus** | |
| i.  X-Direction | 135 GPa |
| ii.  Y/Z-Direction | 9.5 GPa |
| **Poisson's Ratio** | |
| i.  XY | 0.3 |
| ii.  ZY | 0.45 |
| iii.  XZ | 0.3 |
| **Shear Modulus** | |
| i.  XY | 4.90 GPa |
| ii.  ZY | 3.27 GPa |
| iii.  XZ | 4.90 GPa |
| **Orthotropic Thermal Conductivity:** | |
| i.  X-Direction | 5.5 W/(m°C) |
| ii.  Y-Direction | 0.489 W/(m°C) |
| iii.  Z-Direction | 0.658 W/(m°C) |
| Specific Heat, Cp | 1300 W/(m°C) |
| Fiber Volume Fraction | 0.5742 |
| **Resin Properties:** | |
| Initial Degree of Cure | 0.0001 |
| Maximum Degree of Cure | 0.9999 |
| Gelation Degree of Cure | 0.33 |
| Total Heat of Reaction | 540 KJ |
| **Glass Transition Temperature:** | |
| Initial Value | 2.670 °C |
| Final Value | 218.27 °C |
| λ | 0.4708 °C |
| **Orthotropic Cure Shrinkage:** | |
| i.  X-Direction | $1 \times 10^{-20}/mm$ |
| ii.  Y-Direction | 0.0073/mm |
| iii.  Z-Direction | 0.0073/mm |
| **Orthotropic Liquid Pseudo Elasticity:** | |
| i.  X-Direction | 132 GPa |
| ii.  Y-Direction | 165 GPa |
| iii.  Z-Direction | 165 GPa |

*2.2. Simulation Process*

The FEA simulation process was developed using the ACCS package to measure the deformations and residual stresses in the complex composite structure. The ACCS package is a combination of the ANSYS Composite PrepPost (ACP), transient thermal, and static structural analysis modules. The ACCS solver is a crucial tool within the transient thermal module to develop polymerization and find the internal heat generated due to conducted exothermic reactions. It was connected to the structural analysis in this research to study the deformations and shear stresses using the thermal results. ACCS utilizes a fast three-step simulation approach for comparatively thin laminates (<5 mm thick), where an even temperature distribution is assumed [15].

2.2.1. Composite Model Creation

The simulation process starts with the Computer Aided Design (CAD) modeling. In this research, the presented solid model was an L-shaped plate of dimensions 50 × 86 × 1.6 mm and a flange height of 50 mm, as shown in Figure 2a. The analysis was done on a solid composite model to obtain entire thickness cure properties. The design part included 3D solid modeling and .step file conversion in the Creo 4.0 parametric software.

The step file was imported to the ACP module for the composite fiber setup as necessary. The imported solid model's surface was extracted with the ANSYS Spaceclaim platform to create the shell model. A solid model of composite structure with the desired fiber ply thickness, layup angles, and sequence was created using the ACP module. Figure 2b shows the L-plate model generated in the ACP module with the given fiber properties such as 6 layers of Hexcel material oriented by (0/45/90)$_s$. The thicknesses of the fibers used in the model were 0.2 and 0.4 mm.

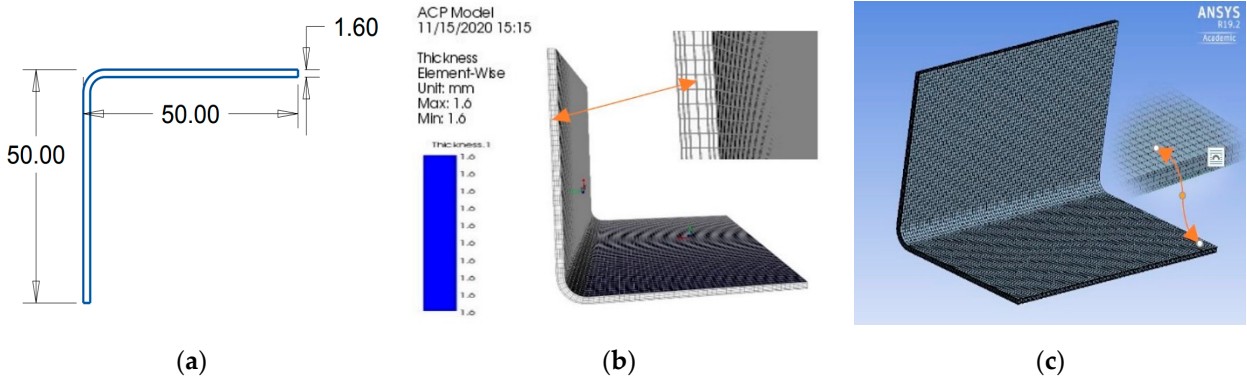

(**a**)         (**b**)         (**c**)

**Figure 2.** Fiber orientation and mesh details in the Advanced Composite PrePost: (**a**) layup details, (**b**) dimensions, and (**c**) mesh details.

Mesh generation was the next significant step in the FEA simulation. Any stress analysis is subject to several types of errors including user error, error due to assumptions and simplifications in the model, and errors due to insufficient mesh discretization. An analysis gives precise results if critical stresses converge to a reasonable level of accuracy. Hemesh Patil [28] presented a mesh convergence study on cylindrical parts. Mesh convergence in a finite elemental study defines the relationship between the number of elements and the accuracy of results. It is necessary to use a desired mesh that is acceptable to the shape and size of the used elements [15]. In the current work, element size was selected to be 1 mm by considering the simulation time and accuracy of results. The sum of observed nodes and elements were 8900 and 8712, respectively. Figure 2c shows the meshing detail for a design structure with a layup of (0/45/90)$_s$. During the simulation, the meshing was generated with smooth quality, including an element size from 0.5 to 2 mm. A significant difference could be observed in the shear stress accuracy and deformation of the results. A comparison of the results with different element sizes is presented in Table 2.

**Table 2.** Significance of mesh size in the deformation.

| Element Size | Nodes | Elements | Deformation | Shear Stress |
|---|---|---|---|---|
| 0.5 | 35,575 | 35,199 | 1.14 | 96.676 |
| 1 | 8900 | 8712 | 4.6191 | 76.893 |
| 1.5 | 4080 | 3953 | 27.968 | 75.213 |
| 2 | 2295 | 2200 | 22.332 | 75.773 |

### 2.2.2. Transient Thermal Analysis

This module's primary purpose is to obtain thermal properties, such as the degree of cure, glass transition temperature, and heat of reaction. This is a three-step non-linear analysis that uses the Newton–Raphson method. A convection condition is given to a composite plate as a thermal input load. The relationship of heat transfer by convection is like the conduction process, which is proportional to the surface area, temperature, and heat transfer coefficient. The rate of reaction in the curing process is proportional to the rate of heat flow [29]. The measured heat generated by the resin during the cure can be

calculated using Equation (1), while the degree-of-cure of the resin can be obtained by integrating the area under the curve of cure rate vs. time, as shown in Equation (2)

$$\frac{d\alpha}{dt} = \frac{1}{H_T}\frac{dH}{dt} \tag{1}$$

$$\alpha = \frac{1}{H_T}\int_0^1 \frac{dH}{dt}dt \tag{2}$$

The convective load resembles the heating process of a composite part. The convection coefficient was taken as 25 W/mm$^2$, and the time step that defined the cure cycle was 240 s, which was kept constant for the full thermal analysis. Two types of cure cycles were used to analyze the thermal behavior of a composite structure. The types of cure cycles and layup sequences have major effects on shape deformation [17,30–32]. Figure 3 shows the double-hold cure cycle in which the temperature rose from 20 to 120 °C and was maintained for 1800 s in the first hold; then, it rose to 180 degrees with a dwell of 1800 s for the second hold; and in the last step, it cooled down to the normal temperature. However, in the single cure cycle as shown in the Figure 4, temperature rose from 20 to 180 degrees and was held for 3600 s [15].

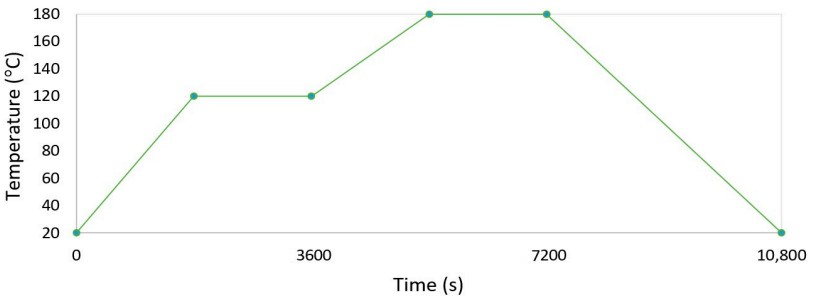

**Figure 3.** Double hold cure cycle.

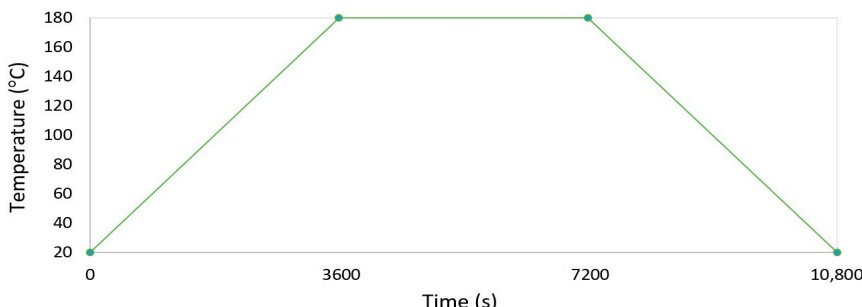

**Figure 4.** Single hold cure cycle.

### 2.2.3. Static Structural Analysis

Structural analysis is a three-step non-linear analysis used to study mechanical behavior like stresses developed in a composite structure. Figure 5 shows a design model with the constraint of given thermal load as an input to get results. The model was constrained at the endpoint of inside curve edges. A fully cured model could provide initial boundary conditions, while frictionless support and remote displacements were the applied load conditions for the structural analysis. The development of residual stresses can also be a function of time [30,31].

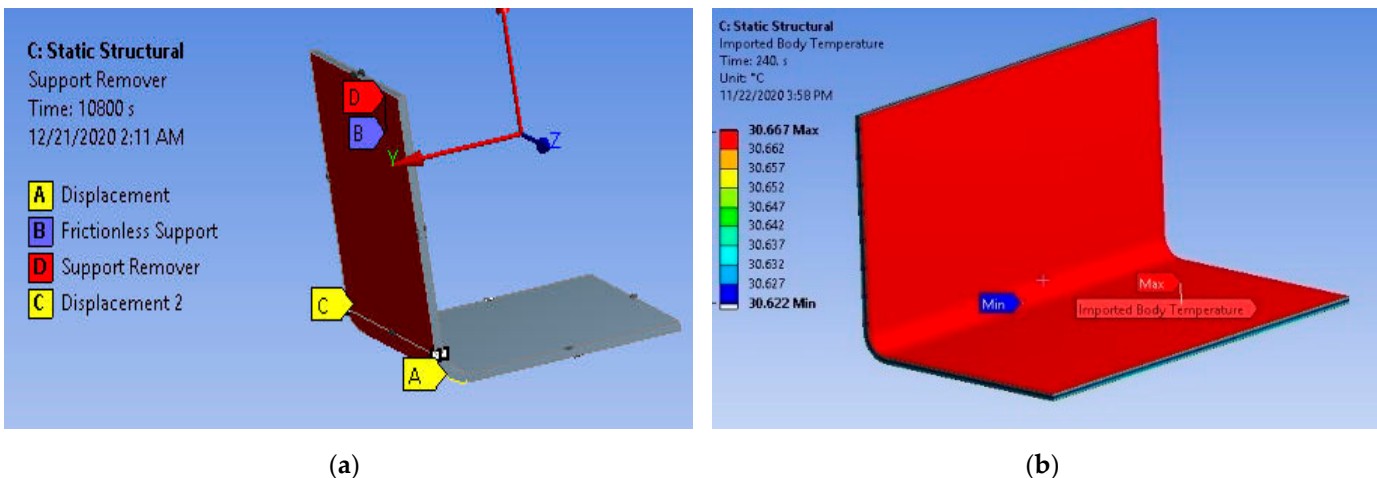

(**a**)　　　　　　　　　　　　　　　　　　　　　　　　(**b**)

**Figure 5.** Static structural analysis model setup: (**a**) Constraints in static structural analysis and (**b**) imported load in static structural analysis.

The stresses in laminate or ply can be calculated from classical laminate theory. Figure 6 explains classical laminate theory in detail. Equation (3) mentions the formula to calculate ply stress [15,32].

$$\left( \begin{array}{c} \sigma_{xx} \\ \sigma_{yy} \\ \tau_{xy} \end{array} \right) = \left( \begin{array}{ccc} Q_{11} & Q_{12} & Q_{16} \\ Q_{12} & Q_{22} & Q_{26} \\ Q_{16} & Q_{26} & Q_{66} \end{array} \right) \left( \begin{array}{c} \varepsilon_{xx} \\ \varepsilon_{yy} \\ \varepsilon_{xy} \end{array} \right) \tag{3}$$

where, $\sigma$, $\tau$, $\varepsilon$, and $Q$ stand for ply stress, shear stress, strain in a single direction, and the transformed reduced stiffness matrix, respectively [33].

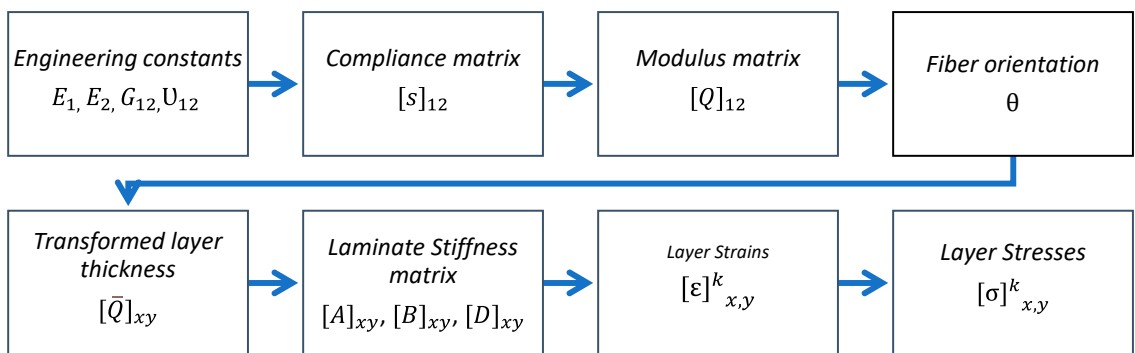

**Figure 6.** Flowchart to find the shear stresses.

### 2.3. Multi-Objective Design Optimization

Design engineers use the optimization technique most of the time to find the parameters that give the best system performance with the desired objectives. This research work presents the basis for a composite structure's optimization by considering limitations in the manufacturing processes and materials. A comprehensive geometrical model is considered for a sample structure to find a precise and useful design with deformation and shear stress. The optimization algorithm is based on the Design and Analysis of Computer Experiments (DACE), in which smart sampling and objectives evaluation drive the design towards a global optimum [23].

Gradient-based and population-based methods can be used in the maximization of the objective function. Gradient-based methods are computationally efficient but local in nature. Population-based methods like GAs and particle swarm optimization (PSO) have more chances to find global solutions for non-linear functions, but they cannot be

guaranteed. The randomly generated population is updated based on the fitness values and random methods until the optima is found.

### 2.3.1. Artificial Neural Network (ANN)

An optimization problem needs realistic boundaries, such as constraints, to keep the result in the desired limit set. Depending on several parameters, such as the number of variables, the solution method, and the used theories, an optimization problem may take a long time. There are several accurate and fast numerical approximation methods to find a given variable function's comparable value. An artificial neural network (ANN) is one of the most accepted ways that is implemented by researchers as a perfect tool to provide fast and reasonable results [27].

An ANN is a biologically inspired computer program formed from hundreds of single units, artificial neurons, or processing elements (PEs) [33]. It comprises several component layers, neurons, and connections. ANNs are trained by detecting patterns and are capable of processing data and making accurate predictions. Figure 7 illustrates the layout of a neural network. 'i' represent the input layer with n parameters specified by the design variables, $h_1$–$h_n$ are hidden layers, and 'o' represents the network's output layer with n objectives. A multi-layer perceptron (MLP) network with three hidden layers and one output layer with two neurons could be defined for the given problem [34–37]. Two neurons were the objectives with the minimum deformation and shear stress.

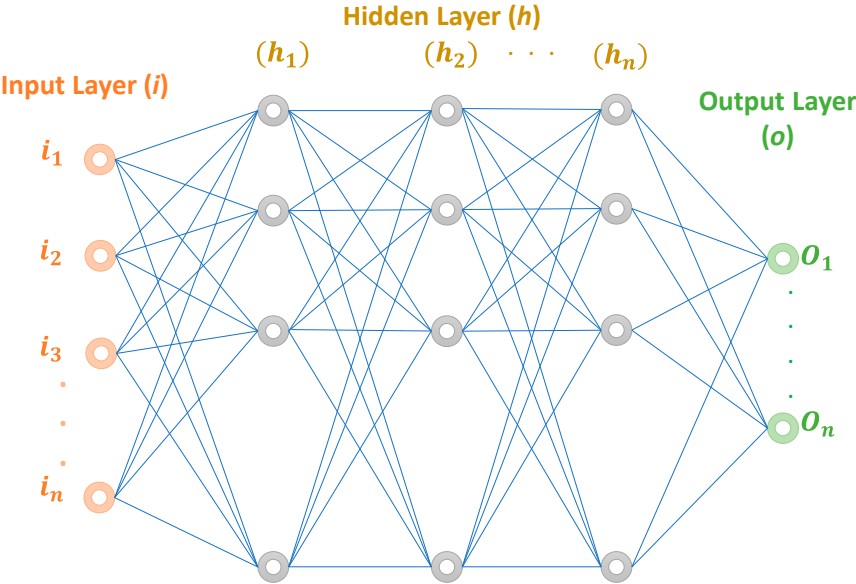

**Figure 7.** The artificial neural network (ANN) structure.

### 2.3.2. Latin Hypercube Sampling

Many design optimization processes start with the selection of design variables. DACE arises when selecting the sample points to be simulated to generate quality results in order to satisfy the objective. It is essential to define the sampling technique that avoids irrelevant simulations. The most frequently used sampling strategies with excellent filling properties are random sampling, stratified sampling, and Latin hypercube sampling (LHS) [34,35]. In the present work, the primary design variable was the layup angle, and the LHS sampling plan was used to generate a sample design variable set. This method ensures that all design space portions are represented in a stratified manner [36]. LHS was used her to generate the initial sample for the layup angles. These variables were then combined with the categorical data [37,38]. The sampling plan used for the project work was as shown in Table 3.

**Table 3.** Latin hypercube sampling plan.

| $\alpha_1$ | $\alpha_2$ | $\alpha_3$ |
|---|---|---|
| 0 | 45 | 90 |
| 0 | −45 | 90 |
| 0 | 0 | 45 |
| 0 | 0 | −45 |
| 0 | 0 | 90 |
| 0 | 45 | 0 |
| 0 | −45 | 0 |
| 0 | 0 | 0 |
| 90 | 90 | 90 |
| 90 | 0 | 45 |

2.3.3. Multi-Objective Optimization Formulation

Multi-objective optimization works as a trade-off analysis because improving one objective implies the worsening of others [2,3,39–42]. In contrary to single-objective optimization problems, generally, there is no unique solution for multi-objective optimization. Therefore, the results of these problems are typically presented by a Pareto frontier curve, which is a set of optimal solutions [27]. A multi-objective optimization problem is defined as:

$$\text{find } x \in \mathbb{R}^{n_{dv}}$$

$$\text{minimize} \quad f(x)$$

$$\text{Subject to } g_i(x) \leq 0, i = 1, 2 \ldots, n_c$$

where $x = [x_1, \ldots \ldots, x_{n_{dv}}]$ is the design vector, $n_{dv}$ is the number of design variables, and f(x) is the vector of the objective functions such as $f(x) = [f_1(x), \ldots., f_k(x)]$, where k is the number of objective functions (output vector). Design constraints can be given by $g_i(x)$, where $n_c$ is the number of constraints.

Many factors affect the amount of residual stress generated in a composite structure during the manufacturing process. A few main ones were considered in this project as design variables. The main input parameters were layup angles and sequences, which were further categorized with the type of cure cycle used and how the material was constrained. A sample number of layup angles considered by using the LHS method is shown in Table 3. In the given design, a total of six layers were taken into consideration. For example, $(0/45/90)_s$ is the structure of the symmetric sequence that was defined as (0/45/90/90/45/0); however, an asymmetric sequence could be defined as (0/45/90/90/−45/0). The present work considered minimum deformation as a primary objective function, and the least value of maximum shear stress generated in the structure was also considered. The design constraints used for the layup angles were limited from −90 to 90. Therefore, the multi-objective optimization statement for the current work is formulated as follows:

$$\text{find } x = [x_1, \ x_2, \ x_3, x_4] \in \mathbb{R}^4$$

$$\text{minimize} \begin{bmatrix} f_1(x_1, \ x_2) \\ f_2(x_1, \ x_2) \end{bmatrix}$$

$$\text{Subject to } g_i(x) = \ \alpha(x_1, x_2) \leq 0$$

$$-90 \leq x_1, x_2 \leq 90$$

## 3. Results and Discussion

### 3.1. Thermal Analysis Results

The convection method implemented during the cure process caused major changes in the resin. At the start, resin was in a viscous flow state. The second stage, which was the resin transition phase, occurred between 3600 and 7200 s and caused the resin to be

violently cured. Its elastic modulus significantly increased, and the resin volume shrank. The structural analysis was performed on the layup with the $(0/45/90)_s$ stacking sequence. The last stage showed that the resin was wholly cured and no chemical reaction took place. Figure 8 shows thermal analysis results for the changes that occurred in the resin transformation phase. It can be seen in the Figure 8 that glass transition temperature changed during the second phase from 3600 to 7200 s, where the heat of reaction (HOR) was at its maximum. Figure 9 shows the temperature distribution in the transient thermal analysis. The temperature was at its maximum at the center and gradually reduced towards the outer ply. Figure 9 also shows the final temperature of the structure after the curing process. The temperature of the structure reduces from 180 to approximately 20 °C, with a final temperature of 20.044 °C. The glass transition temperature $T_g$ significantly affected the resin mechanical properties, changing it from a rubbery state to a glassy state [15].

$$\frac{T_g - T_{g0}}{T_{g\infty} - T_{g0}} = \frac{\lambda_\alpha}{1 - (1 - \lambda)\alpha} \tag{4}$$

where $T_g$ is the glass transition temperature; $T_{g\infty}$ and $T_{g0}$ are the glass transition temperatures of uncured and fully cured resin, respectively; and $\alpha$ is the degree-of-cure.

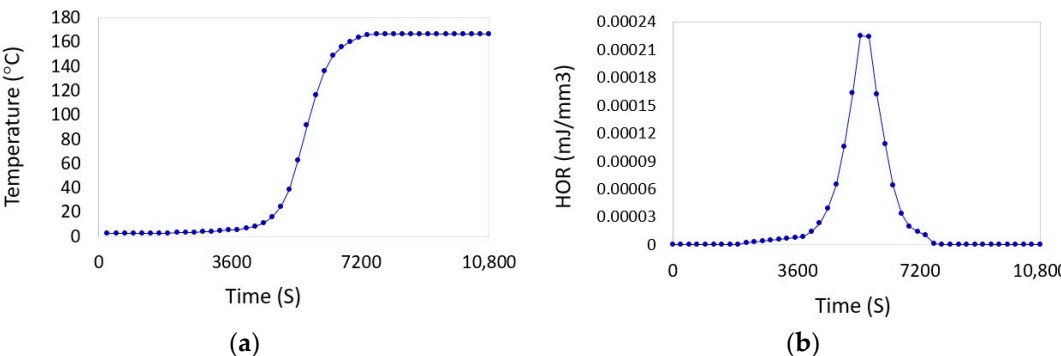

(a)  (b)

**Figure 8.** Resin transformation in the thermal analysis: (**a**) glass transition temperature and (**b**) heat of reaction (HOR).

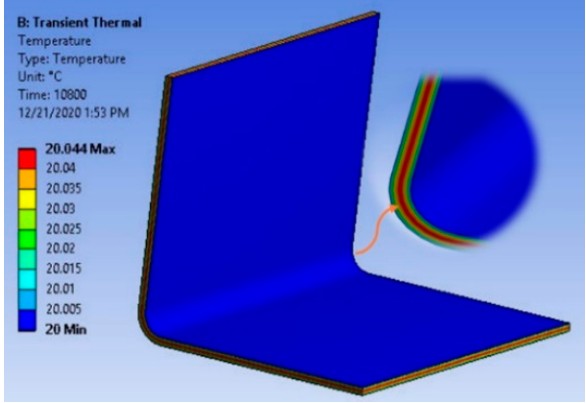

**Figure 9.** Temperature distribution in thermal analysis.

From the results of the thermal analysis, it was possible to calculate the instantaneous composite elastic constants. Similarly, the micromechanics approach could be used to calculate thermal and chemical strains for a given increment [37].

### 3.2. Static Structural Results

Static structural analysis was the final step implemented to calculate actual deformations and residual stresses. In the composite structure, the fibers created strength and stiffness, while the matrix provided bonding. Thus, the composite possessed good me-

chanical properties parallel to the fibers and was relatively weak in the perpendicular direction.

The structural analysis was performed on the layup with the $(0/45/90)_s$ stacking sequence. Figure 10 shows a graphical representation of the results based on the maximum shear stress and deformations. Figure 10a shows a graphical representation of the maximum amount of shear stress that could be generated in the structure. For the given diagram, the amount of shear stress generated after curing was 76.893 MPa. Figure 10b shows the total deformation that occurred in the same structure due to the generated residual stresses, and it can be seen in the diagram that the deformation occurred at up to 4.6191 mm for the layup.

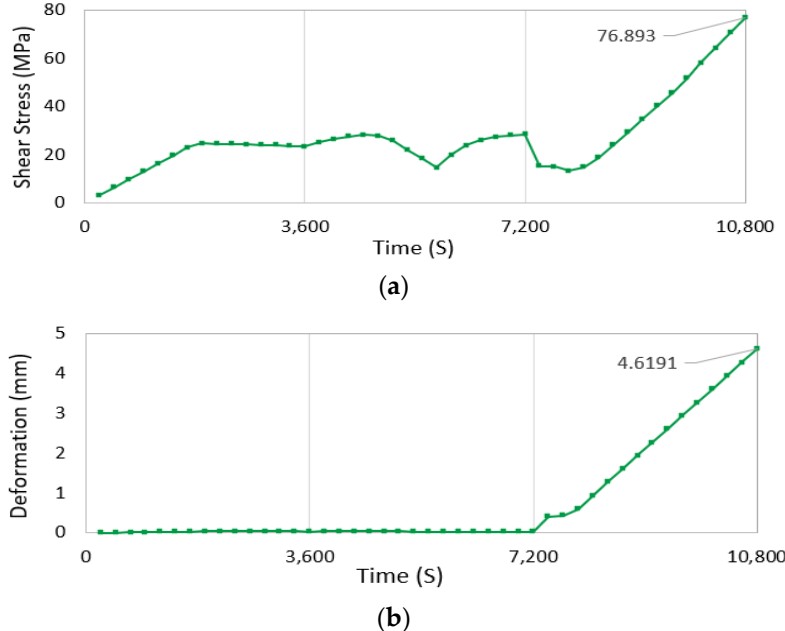

**Figure 10.** Graphical representation of results of stress analysis: (**a**) maximum shear stress and (**b**) total deformation.

Figure 11a,b shows the stress analysis results with spring-in, twist, and warpage. If a complex composite structure is investigated using the ACCS tool, bending, sagging, and/or twisting may occur (as Figure 11c shows this might behappened in similar studies), which could be deleterious for the final component's life.

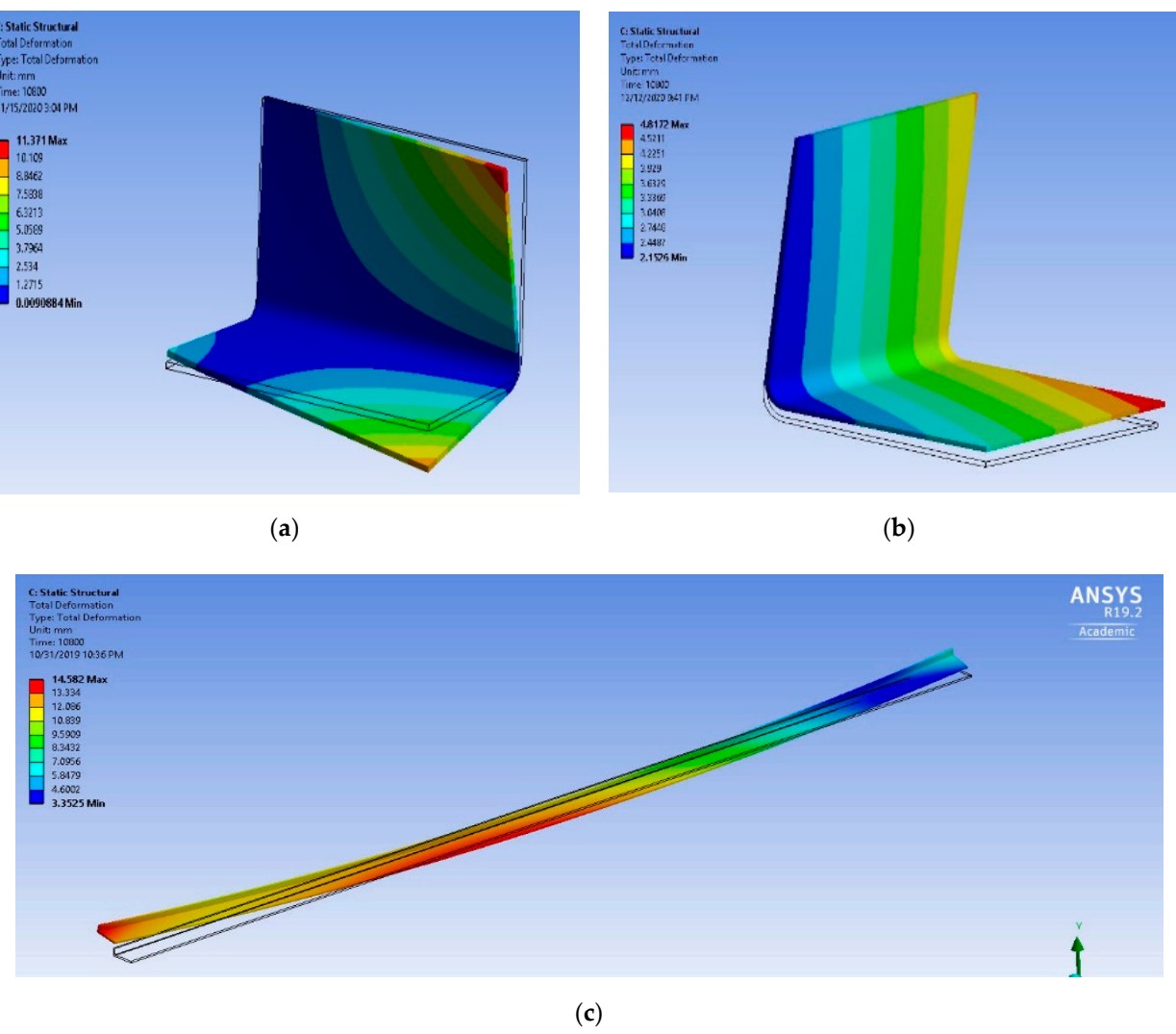

**Figure 11.** Possible deformations in the structure in the curing process: (**a**) warpage, (**b**) spring-in, and (**c**) sagging/bending.

### 3.3. Parameter Study Results

The workflow diagram shown in Figure 12 explains the network flow of the optimization process with the categories included in the process. It started with sample layup angles decided with the LHS plan. The selected α, i.e., the values of different layup angles, were further divided into several categories such as symmetry, cure cycle, and constraints. All the design parameters were then studied to analyze the output as minimum deformation and shear stress in the component. The sample data collected from the simulations were saved into the spreadsheet and are explained with a comparison to show the optimized structure. Table 4 shows the values of all the simulation results based on all the input design variables. The same data were further used in the scatter diagram, as shown in Figure 13, to represent the results [43–47].

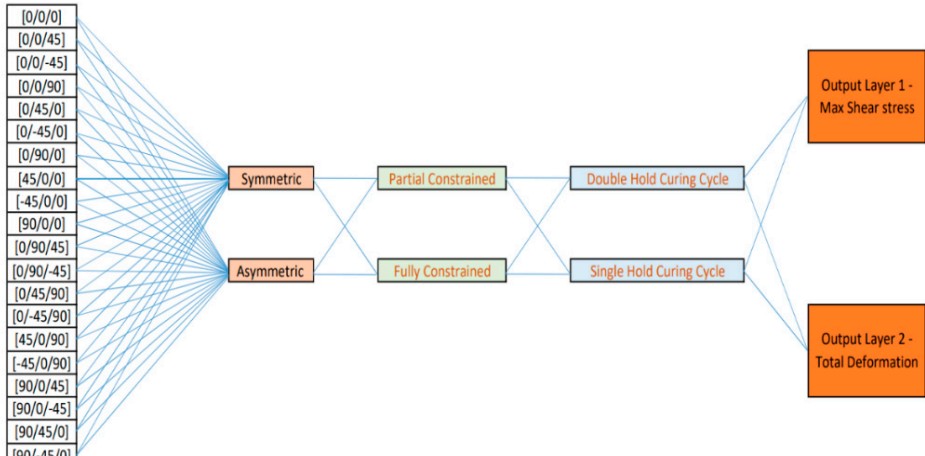

**Figure 12.** The details of categories that can be considered to build the neural network.

**Table 4.** Simulation results (Srs).

| Sr No | Design Variable 1 | | | Design Variable 2 | Design Variable 3 | Partially Constrained | | Fully Constrained | |
|---|---|---|---|---|---|---|---|---|---|
| Design | a1 | a2 | a3 | Symmetry | Cure Cycles | Deformations | Max Shear Stress | Deformations | Max Shear Stress |
| D1 | 0 | 45 | 90 | Symmetric | Single | 20.605 | 82.662 | 0.585062 | 87.257 |
| D2 | 0 | 45 | 90 | Symmetric | Double | 4.6191 | 76.893 | 0.52853 | 96.252 |
| D3 | 0 | 45 | 90 | Asymmetric | Single | 14.35 | 100.78 | 1.9054 | 161.43 |
| D4 | 0 | 45 | 90 | Asymmetric | Double | 3.9376 | 93.543 | 1.7711 | 149.03 |
| D5 | 0 | −45 | 90 | Symmetric | Single | 6.7752 | 83.235 | 0.58509 | 87.833 |
| D6 | 0 | −45 | 90 | Symmetric | Double | 0.62761 | 75.102 | 0.54317 | 82.168 |
| D7 | 0 | −45 | 90 | Asymmetric | Single | 17.044 | 77.022 | 2.1651 | 190.16 |
| D8 | 0 | −45 | 90 | Asymmetric | Double | 3.5887 | 72.327 | 2.0126 | 176.04 |
| D9 | 0 | 0 | 45 | Symmetric | Single | 22.091 | 460.74 | 0.64866 | 458.43 |
| D10 | 0 | 0 | 45 | Symmetric | Double | 0.62892 | 428.89 | 0.61548 | 425.63 |
| D11 | 0 | 0 | 45 | Asymmetric | Single | 16.951 | 388.01 | 4.7133 | 400.26 |
| D12 | 0 | 0 | 45 | Asymmetric | Double | 9.1218 | 360.25 | 4.4283 | 371.67 |
| D13 | 0 | 0 | −45 | Symmetric | Single | 7.0188 | 458.4 | 0.6489 | 457.54 |
| D14 | 0 | 0 | −45 | Symmetric | Double | 2.2174 | 431.04 | 0.6297 | 425.68 |
| D15 | 0 | 0 | −45 | Asymmetric | Single | 51.262 | 387.19 | 4.7137 | 401.24 |
| D16 | 0 | 0 | −45 | Asymmetric | Double | 11.371 | 359.57 | 4.5054 | 372.67 |
| D17 | 0 | 0 | 90 | Symmetric | Single | 2.5907 | 8.232 | 0.59803 | 68.097 |
| D18 | 0 | 0 | 90 | Symmetric | Double | 1.5114 | 63.335 | 0.55558 | 63.197 |
| D19 | 0 | 0 | 90 | Asymmetric | Single | 18.757 | 95.12 | 0.62251 | 95.06 |
| D20 | 0 | 0 | 90 | Asymmetric | Double | 2.5089 | 88.294 | 0.57862 | 88.216 |
| D21 | 0 | 45 | 0 | Symmetric | Single | 41.052 | 22.741 | 0.63468 | 552.69 |
| D22 | 0 | 45 | 0 | Symmetric | Double | 17.676 | 558.9 | 3.3133 | 20.216 |
| D23 | 0 | 45 | 0 | Asymmetric | Single | 12.784 | 575.29 | 5.4332 | 598.99 |
| D24 | 0 | 45 | 0 | Asymmetric | Double | 15.849 | 536.4 | 5.1348 | 558.54 |
| D25 | 0 | −45 | 0 | Symmetric | Single | 15.58 | 599.15 | 0.6342 | 552.34 |
| D26 | 0 | −45 | 0 | Symmetric | Double | 16.522 | 557.7 | 0.60208 | 513.34 |
| D27 | 0 | −45 | 0 | Asymmetric | Single | 33.833 | 143.38 | 5.4338 | 596.4 |
| D28 | 0 | −45 | 0 | Asymmetric | Double | 11.343 | 573.84 | 29.024 | 31.582 |
| D29 | 0 | 0 | 0 | Symmetric | Single | 111.51 | 562.62 | 0.63454 | 557.98 |
| D30 | 0 | 0 | 0 | Symmetric | Double | 1.4481 | 522.28 | 0.59507 | 518.04 |

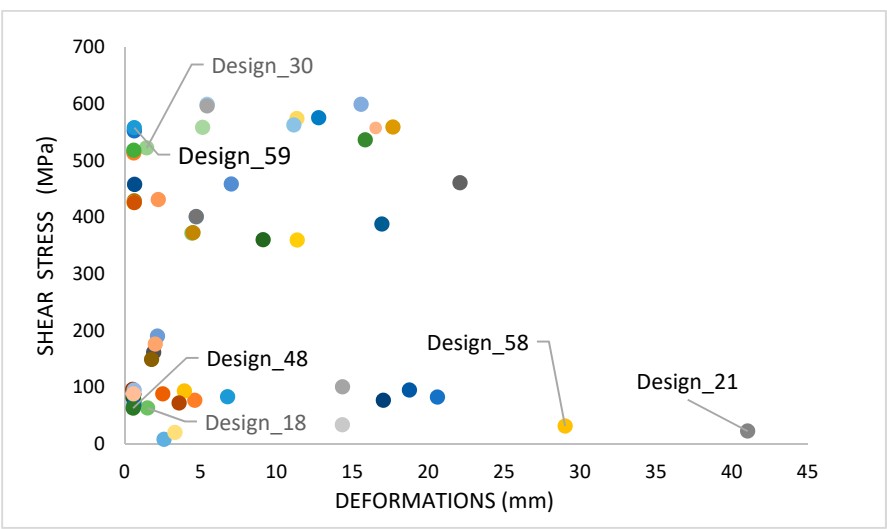

**Figure 13.** Scatter diagram showing a comparison of the ACCS results.

A graphical representation of all simulation results is shown in Figure 14. The relation between both objective functions is explained in the graph. The deformation values are on the *X*-axis, and shear stress is on the *Y*-axis. The values of all results are shown in Table 5. A feasible objective space defined in the graph illustrates the possible solutions for all design variables. It can be seen from the graph that few samples like Design_30 and Design_59 could meet the first objective, minimum deformation, but the value of shear stress-induced deformation was higher. Similarly, Design_21 and Design_58 had less shear stress, but they did not meet the objective of minimum deformation. Such designs could not be concluded as an optimum design. As mentioned by Ehsani, A., and H. Dalir [27] in their research, multi-objective optimization works on a trade-off analysis. Therefore, the result of these problems is typically presented by a curve with a set of optimal solutions known as a Pareto frontier. In this research work, all the design solutions had minimum deformations arranged with the given shear stress. As seen in the given figure, Design_48 and Design_18 showed the best results that could satisfy both the objectives of minimum deformation and shear stress.

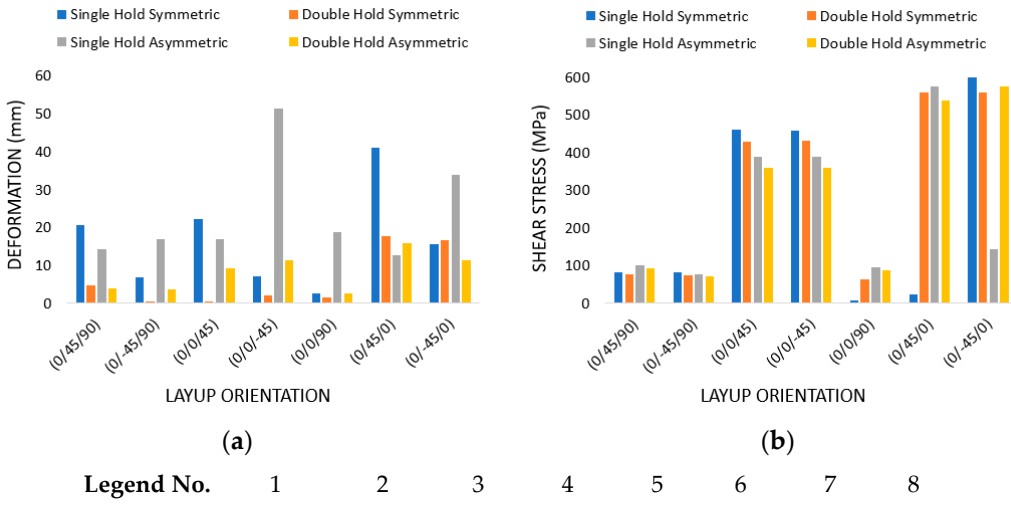

(**a**)                                      (**b**)

| **Legend No.** | 1 | 2 | 3 | 4 | 5 | 6 | 7 | 8 |
|---|---|---|---|---|---|---|---|---|

**Layup Detail** (0/45/90) (0/−45/90) (0/0/45) (0/0/−45) (0/0/90) (0/45/0) (0/−45/0) (0/0/0)

**Figure 14.** Comparison of residual stresses based on symmetric and asymmetric layup: (**a**) shear stress and (**b**) deformation.

**Table 5.** Deformation and shear stress of all designs.

| Design_1 | Design_2 | Design_3 | Design_4 | Design_5 | Design_6 | Design_7 | Design_8 | Design_9 | Design_10 | Design_11 | Design_12 | Design_13 | Design_14 | Design_15 |
|---|---|---|---|---|---|---|---|---|---|---|---|---|---|---|
| 20.605 | 4.6191 | 14.35 | 3.9376 | 6.7752 | 0.62761 | 17.044 | 3.5887 | 22.091 | 0.62892 | 16.951 | 9.1218 | 7.0188 | 2.2174 | 51.262 |
| 82.662 | 76.893 | 100.78 | 93.543 | 83.235 | 75.102 | 77.022 | 72.327 | 460.74 | 428.89 | 388.01 | 360.25 | 458.4 | 431.04 | 387.19 |
| **Design_16** | **Design_17** | **Design_18** | **Design_19** | **Design_20** | **Design_21** | **Design_22** | **Design_23** | **Design_24** | **Design_25** | **Design_26** | **Design_27** | **Design_28** | **Design_29** | **Design_30** |
| 11.371 | 2.5907 | 1.5114 | 18.757 | 2.5089 | 41.052 | 17.676 | 12.784 | 15.849 | 15.58 | 16.522 | 33.833 | 11.343 | 111.51 | 1.4481 |
| 359.57 | 82.32 | 63.335 | 95.12 | 88.294 | 22.741 | 558.9 | 575.29 | 536.4 | 599.15 | 557.7 | 143.38 | 573.84 | 562.62 | 522.28 |
| **Design_31** | **Design_32** | **Design_33** | **Design_34** | **Design_35** | **Design_36** | **Design_37** | **Design_38** | **Design_39** | **Design_40** | **Design_41** | **Design_42** | **Design_43** | **Design_44** | **Design_45** |
| 0.585062 | 0.52853 | 1.9054 | 1.7711 | 0.58509 | 0.54317 | 2.1651 | 2.0126 | 0.64866 | 0.61548 | 4.7133 | 4.4283 | 0.6489 | 0.6297 | 4.7137 |
| 87.257 | 96.252 | 161.43 | 149.03 | 87.833 | 82.168 | 190.16 | 176.04 | 458.43 | 425.63 | 400.26 | 371.67 | 457.54 | 425.68 | 401.24 |
| **Design_46** | **Design_47** | **Design_48** | **Design_49** | **Design_50** | **Design_51** | **Design_52** | **Design_53** | **Design_54** | **Design_55** | **Design_56** | **Design_57** | **Design_58** | **Design_59** | **Design_60** |
| 4.5054 | 0.59803 | 0.55558 | 0.62251 | 0.57862 | 0.63468 | 3.3133 | 5.4332 | 5.1348 | 0.6342 | 0.60208 | 5.4338 | 29.024 | 0.63454 | 0.59507 |
| 372.67 | 68.097 | 63.197 | 95.06 | 88.216 | 552.69 | 20.216 | 598.99 | 558.54 | 552.34 | 513.34 | 596.4 | 31.582 | 557.98 | 518.04 |

Figure 15 shows the clustered column plot of results based on the curing cycle for different orientations. Figure 15a explains the difference in the deformation results that occurred with specific layup designs having single hold and double hold curing cycles. It can be seen from the graph that the single hold process could have more deformation in the structure than the double hold cure cycle. Similar plots are provided for the shear stress comparison. In the single hold cure cycle, the structure was held at 180 degrees for an hour, which resulted in more deformations than in the double hold cycle, where the structure could cure in two steps at 120 and 180 degrees for 30 min. Details of the deformation and shear stress data are shown in Table 6.

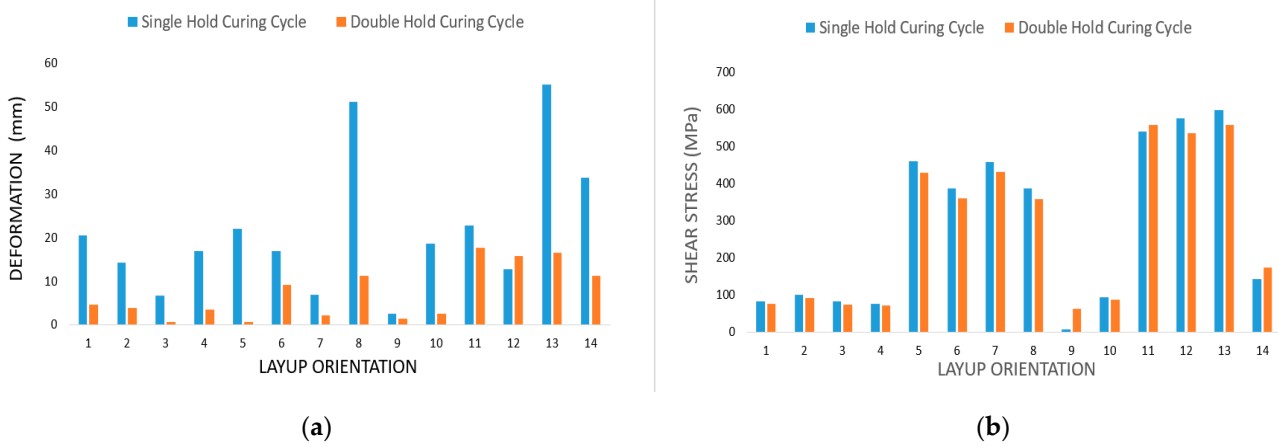

**(a)**                                                                 **(b)**

**Figure 15.** Scatter diagram showing a comparison of results based on the curing cycle: (**a**) deformation and (**b**) shear stress.

The double cure cycle showed better results than the single cure cycle because of the resin-induced effects. Literature on cure cycling implied that if the curing reaction of the resin occurred too quickly, the resin flow time would be reduced and result in voids and deformations. It is essential to ensure the complete cure of final laminates, which guarantees a good laminate quality [19]. Dong [18] presented a study of laminate quality based on the initial cure temperature and cure cycle. They mentioned that the degree of cure increased with cure time until reaching a constant value. For our study in the double hold cure cycle, the maximum degree of cure at 120 °C was 0.78, which suggested an insufficient cross-linking network formation. When the temperature rose to 180 °C, the degree of cure (DOC) reached 0.95, which resulted in a better laminate quality with less deformation. Additionally, the porosity in the laminate was at a minimum for the double hold cure cycle with dwelling at 120 and 180 °C.

**Table 6.** ACCS results provided to show a curing cycle comparison.

| Layup Orientation | Deformation | | Shear Stress | |
|---|---|---|---|---|
| | **Single Hold** | **Double Hold** | **Single Hold** | **Double Hold** |
| $[0/45/90]_s$ | 20.605 | 4.6191 | 82.662 | 76.893 |
| $[0/45/90]_{as}$ | 14.35 | 3.9376 | 100.78 | 93.543 |
| $[0/-45/90]_s$ | 6.7752 | 0.62761 | 83.235 | 75.102 |
| $[0/-45/90]_{as}$ | 17.044 | 3.5887 | 77.022 | 72.327 |
| $[0/0/45]_s$ | 22.091 | 0.62892 | 460.74 | 428.89 |
| $[0/0/45]_{as}$ | 16.951 | 9.1218 | 388.01 | 360.25 |
| $[0/0/-45]_s$ | 7.0188 | 2.2174 | 458.4 | 431.04 |
| $[0/0/-45]_{as}$ | 51.262 | 11.371 | 387.19 | 359.57 |
| $[0/0/90]_s$ | 2.5907 | 1.5114 | 8.232 | 63.335 |
| $[0/0/90]_{as}$ | 18.757 | 2.5089 | 95.12 | 88.294 |
| $[0/45/0]_s$ | 22.741 | 17.676 | 541.052 | 558.9 |
| $[0/45/0]_{as}$ | 12.784 | 15.849 | 575.29 | 536.4 |
| $[0/-45/0]_s$ | 55.2 | 16.522 | 599.15 | 557.7 |
| $[0/-45/0]_{as}$ | 33.833 | 11.343 | 143.38 | 573.84 |

All the design parameters selected in the process generated different deformations and shear stresses. A few of the common trends are shown in Figure 16, which shows results based on the symmetry and curing process for the given layup orientation. The graph demonstrates that the symmetric layup cured with a single hold cycle could generate more shear stress. The fifth layup with the $[0/0/90]_s$ orientation gave the best results for the shear stress. However, deformation results were diverse among the optimum cases. Moheimani et al. presented an important approach to study the failure of epoxy laminae using a cohesive multiscale model [38].

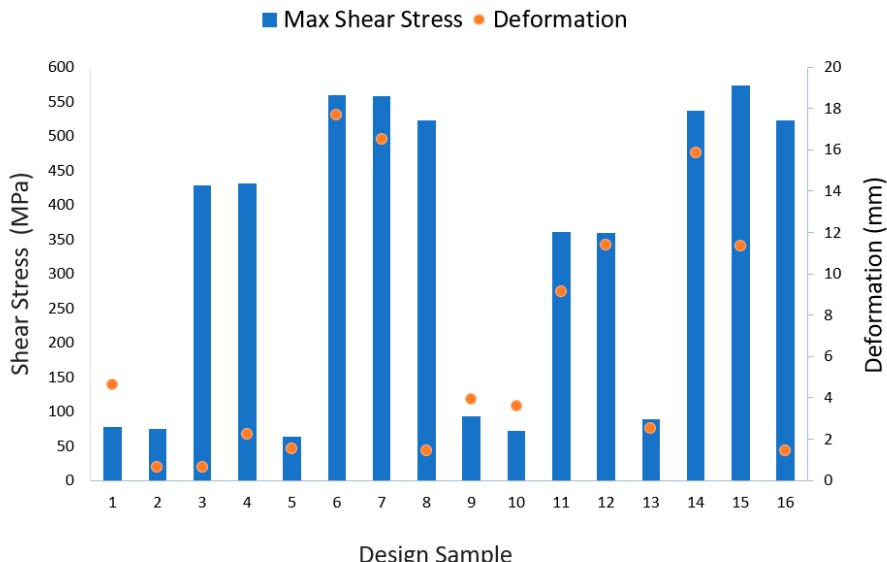

**Figure 16.** Layup angle comparison with reference to deformation and shear stress.

Figure 16 shows a combination bar and scatter graph, indicating the stress analysis results for the given layup orientation samples. The *X*-axis describes all the layup orientation samples considered in the process. The left *Y*-axis is used to measure the deformation results, while the right vertical axis is for the shear stresses. The bar graph combined with the scatter diagram shows the variations in the maximum shear stress (MPa) and deformation (mm).. The graph shows that the best results can be found in some layup orientation samples like design 5, 9, 10 and 13. These samples were considered in the feasible design space.

### 3.4. Validation of Simulation Results with Experiment

A comparison of the ACCS results with the experimental one was reviewed. It was found that few researchers have presented work on a comparison of deformation results. T. Garstka mentioned that they have done experimentation to validate spring-in results [39,40]. In the current work, a sample layup with the stacking sequence of $[0/45/90]_s$ was used to compare spring-in values based on analytical, numerical, and experimental procedures. The results showed that the analytical calculations gave a 0.90° spring-in. However, the ACCS simulation and experimental results showed spring-in deformations of 0.81° and 0.78°, respectively. It can be observed from the results that the simulation and experimental results did not have significant differences in their deformation values [11].

### 4. Conclusions

In the current research, the ACCS tool was utilized to generate data with different references such as the curing cycle, constraints, and layup symmetry, which were further investigated to find the best results after manufacturing. The fundamentals of the multi-objective optimization of a complex composite structure were described while considering two objective functions: the least amount of deformation and the shear stress generated during the thermal analysis. The process started with the DOE, in which the LHS sampling method was developed to generate the samples of fiber orientation angle. The design variables were categorized according to the used curing process, constraints, and stacking sequence. These results can be used to compensate for tool design to reduce errors in a prototype. The ACCS result data were analyzed to show a comparison based on the layup orientation.

The results signified that the curing process and the layup sequence used in the composites changed the deformation results and the amount of shear stress generated during cure. The data found with the ACCS tool led us to conclude that the single hold curing cycle caused more deformation than the double hold cycle. Furthermore, the scatter diagram of all studied cases showed that designs 33, 34, 38, and 46 had the best performance after manufacturing. These designs can be accommodated in a feasible design space. On the contrary, the samples made with designs 41 and 63 were found to be outside of the feasible design space; hence, they can be considered as non-feasible points. The comparison of layup orientations as shown in Figure 16 implied that the 5th $((0/0/45)_s)$ and 13th $((0/-45/0)_s)$ layup sequences had the most optimized results. A good agreement was obtained between with the empirical case study and the simulation modeling. We can claim that the foundation of this study like input and output parameters can be implemented in an ANN to find an optimal structure. In other words, the data executed with the FEA tool can be utilized to train parametric data in an artificial neural network.

**Author Contributions:** Methodology, software, formal analysis, and writing/draft preparation—N.K., methodology, review, supervision, and writing/editing—R.M., project supervision and administration—H.D. All authors have read and agreed to the published version of the manuscript.

**Funding:** The authors would like to express their gratitude to Indy Car (#073779) for the funding provided to do this research and for their assistance with the instrumentation and proofreading.

**Acknowledgments:** Authors would like to thank Simutech for their valuable support with the software tutorial and license. We would like to acknowledge partial funding from Indy Car Group for supporting this research work.

**Conflicts of Interest:** The authors declare no conflict of interest.

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
