# Peer review of "Analysis of Composite Structures in Curing Process for Shape Deformations and Shear Stress: Basis for Advanced Optimization"

_jcs, doi:10.3390/jcs5020063_

Round 1
Reviewer 1 Report
The paper claims to be dealing with "the optimization of composite structures in curing process", an topic of high practical relevance. The topic is clearly within the scope of the journal.
Unfortunately to the reviewer neither the optimization nor the use of the ANN (as stated in the introduction) are clearly presented. In the opinion of the reviewer, a mayor revision is necessary before the article can be considered for publication
My comments in detail:
Figure 1:
Please check, if all the arrows are needed | correct. In the eyes of the reviewer at least the bold arrows between "Execute FEA" => " Evaluate the objective function" => "Finalize the design" are irritating
Table 1:
Please add information, that X-direction is equivalent to fibre direction or 0°-direction of the UD-prepreg
Section 2.2.1:
The authors are mentioning the necessity of a mesh convergence study. It is unclear, whether a mesh convergency study has been carried out for the model dealt with in this study or not.
Figure 4:
Please add ordinate axis labelling
Line 213 (Equation 4)
- Please format the symbols used in the equation using subscripts.
- In the case studies here, are the stresses and strains constant over the ply thickness or can bending effects be observed, that have to be taken into account?
Line 232:
"can find global solutions" It is true, that population-based optimization methods have a greater chance to find a global optimum. But this is not guaranteed! Please add a short comment on this fact.
Line 292:
The objective function is a vector. So what is meant by "minimize"? Should the components of the vector be minimized separately? Or what mathematical norm is used?
Figure 7 and describing text (line 246 to 250):
The text and the figure are not consistent. In the figure n-Input neurons are shown, according to the text there should be 4, the same is for the output layer. B1 to b3 are not shown in the figure. The number of neurons in the hidden layer and the used activation functions are not given.
Line 330:
"at the given layup" which one was used? As in section 2.3.1 LHS is described and a table of different layups in given, it is unclear, for which layup the results are presented.
Line 363 to 373:
The nomenclature "desing-26" etc has not been introduced. So it is unclear what this expression stands for. An explanation is necessary!
Figure 11:
Caption is missing
Figure 14:
On the x-axis the different layup-configurations are referenced. These are only discrete points and the values are not continuously distributed. Therefore in the opinion of the reviewer it is not possible to generate a graph by connecting the y-values with a line.
Table 4:
Not referenced in the text.
Conclusion:
The authors claim, that an optimization using an ANN is presented. For the reviewer it is uncertain, where in the process an optimization takes place and where the ANN is used. Neither the training nor the testing of the ANN are described. Therefore neither the sensitivity nor the specificity of the ANN can be checked. Looking at the low number of simulation parameter sets described in the paper, in the eyes of the reviewer the questions arises, if an ANN trained on such a low number of datasets is really working. Before publication the set-up, training and test of the ANN must be described and an estimation on sensitivity and specificity must be provided. Furthermore, the authors must point out more clearly, where the ANN is used in the optimization procedure and where the optimization takes place.
At this state of the publication, the reviewer can see a compilation of different results which (perhaps by chance) show some good guesses but no optimization. The use of the ANN is unclear. The use of the optimization is unclear.
Author Response
- Figure1: Please check, if all the arrows are needed | correct. In the eyes of the reviewer at least the bold arrows between "Execute FEA" => "Evaluate the objective function" => "Finalize the design" are irritating
Answer:
We thank the editor for the comment. Figure 1 is modified as suggested.
- Table1:
Please add information, that X-direction is equivalent to fibre direction or 0°-direction of the UD-prepreg
Answer:
As per suggestion, information added in the section 2.1 materials. ‘The X and Y direction indicates the fiber direction as parallel (00) and Transverse (900) to the fiber’
- Section 2.2.1:
The authors are mentioning the necessity of a mesh convergence study. It is unclear, whether a mesh convergence study has been carried out for the model dealt with in this study or not.
Answer:
Authors are thankful to the reviewer for this very constructive suggestion. We tried to show the significance of the mesh in the FEA. We referred to Hemesh's [29] and Ameya’s [11] work to describe that the mesh quality values are vital and directly linked with the accuracy of results. For the given problem, we did a tarde-off to select element size 1 mm as time/energy-efficient case. The number of nodes and elements were 8900 and 8712. It has been observed that the size we considered predicts accurate results with a lower simulation time with respect to others. The details of comparative study are discussed in the table 2. of section 2.2.1. Table below investigates the L-plate model generated in the ACP module with the given fiber properties such as 6 layers of Hexcel material oriented by [0/45/90]s.
Table 2. Significance of mesh size in the deformation
|
Element Size (mm) |
Nodes |
Elements |
Deformation angle 0 |
Shear Stress (MPa) |
|
0.5 |
35575 |
35199 |
1.14 |
96.676 |
|
1 |
8900 |
8712 |
4.6191 |
76.893 |
|
1.5 |
4080 |
3953 |
27.968 |
75.213 |
|
2 |
2295 |
2200 |
22.332 |
75.773 |
- Figure 4: Please add ordinate axis labelling
Answer:
Thank you for pointing this out, Figure 4 replaced by the new one with ordinate axis labelling and unit.
- Line 213 (Equation 3)
- Please format the symbols used in the equation using subscripts. In the case studies here, are the stresses and strains constant over the ply thickness or can bending effects be observed, that have to be taken into account?
Answer:
The format of the equation has been changed as suggested. The new one is shown on line #221. For the current design, we have considered constant stress over the ply without considering the bending effects. However, significant bending effects may occur in the long and complex structure. We mentioned in line #344 to show bending effects may occur in complex structures (figure 11c). We are working on the study of using the ACCS tool for the analysis of such complex composite structures. We will investigate this data in future work.
- Line 232:
"can find global solutions" It is true, that population-based optimization methods have a greater chance to find a global optimum. But this is not guaranteed! Please add a short comment on this fact.
Answer:
Thank you for this information. The sentence is paraphrased by adding the new comment as suggested. It is mentioned on line #237.
- Line 292:
The objective function is a vector. So what is meant by "minimize"? Should the components of the vector be minimized separately? Or what mathematical norm is used?
Answer:
Yes, they (both objectives deformation and shear stress) need to get minimized separately. In this project work, we considered the optimization procedure, in which the group of low deformation results used to search the feasible objective space. These designs with lower deformations are considered to run further simulations to find the minimal value of shear stresses. As mentioned in the results, a few designs satisfy one objective as least deformation but show higher shear stresses and vice versa. Figure 13 shows the details of all results with a scatter plot.
- Figure 7 and describing text (line 246 to 250):
The text and the figure are not consistent. In the figure n-Input neurons are shown, according to the text there should be 4, the same is for the output layer. B1 to b3 are not shown in the figure. The number of neurons in the hidden layer and the used activation functions are not given.
Answer:
We thank the reviewer for a detailed review of our manuscript. The changes implemented in the description of the figure. Figure 7 shows the general neural network with n number of layers. We have edited the content accordingly. The corrected data can be found in line #252
- Line 330:
"at the given layup" which one was used? As in section 2.3.1 LHS is described and a table of different layups in given, it is unclear, for which layup the results are presented.
Answer:
We thank the reviewer for raising this point. We have developed a MATLAB code to get the sample values using the Latin Hypercube sampling plan. It generates random results every time we run the code. We have considered the same sample layups generated by the LHS algorithm as shown in table 3 to run the simulations. Table 4 shows the simulation results for the layups that are generated by the LHS plan, as shown in table 3.
Line 363 to 373:
The nomenclature "desing-26" etc has not been introduced. So it is unclear what this expression stands for. An explanation is necessary!
Answer:
With thanks, the table of all the detail has been added below the scatter plot. It can be seen in the Table 5 on line number #384
- Figure 11: Caption is missing
Answer:
Caption has been added in the line #350. (Figure 11. Possible deformations in the complex structure after curing process (a) warpage (b) Spring-in (c) sagging/bending.
- Figure 14:
On the x-axis the different layup-configurations are referenced. These are only discrete points and the values are not continuously distributed. Therefore in the opinion of the reviewer it is not possible to generate a graph by connecting the y-values with a line.
Answer:
We do agree with your point. Since there is no continuation in deformations results and shear stress, values on the y axis should not be interconnected. The graph has changed to the column bar graph, which compares the single hold VS double hold cure cycle. Because of this change, there will be no line connecting to all the y-values in the graph.
- Table 4: Not referenced in the text.
Answer:
Thanks for your meticulous attention. Table number has been now changed to table 6 and it contains the supporting data for the graph which is referred on the line #395.
Conclusion:
The authors claim, that an optimization using an ANN is presented. For the reviewer it is uncertain, where in the process an optimization takes place and where the ANN is used. Neither the training nor the testing of the ANN are described. Therefore neither the sensitivity nor the specificity of the ANN can be checked. Looking at the low number of simulation parameter sets described in the paper, in the eyes of the reviewer the questions arises, if an ANN trained on such a low number of datasets is really working. Before publication the set-up, training and test of the ANN must be described and an estimation on sensitivity and specificity must be provided. Furthermore, the authors must point out more clearly, where the ANN is used in the optimization procedure and where the optimization takes place. At this state of the publication, the reviewer can see a compilation of different results which (perhaps by chance) show some good guesses but no optimization. The use of the ANN is unclear. The use of the optimization is unclear.
Answer:
We are grateful to the reviewer for the meticulous observation. This paper intends to the comparison of the ACCS results. We reproduce the results mentioned in the Amey Patil [11,46-47] research work with the detailed study by doing an experiment work as a case study. Our current work represents a strong foundation study on an innovative method which is developed to find the residual stresses.
We have not trained the algorithm as yet. We would investigate the results by using the Genetic Algorithm (GA) in the future, in which we will also implement an artificial neural network (ANN). We have added some contents about the ANN to show its significance in the optimization process and to show that we can develop an algorithm based on the results found in our study. However, the data (line #351) has been paraphrased with better content. Considering the content added, we are not claiming that we used ANN. Instead, we focused on the comparison and stated that the foundation of this study and input and output parameters are ready to get implemented in ANN to find the optimal structure. In another word, the data executed in the FEA tool would be utilized to train the parametric data in artificial neural network.

Reviewer 2 Report
This work demonstrates an application of ANSYS Composite Cure Simulation (ACCS) tool to simulate deformations and residual stresses induced in the composite structures due to the various design parameters used in manufacturing. Overall, the objectives of the paper are clear, and the methods and results are presented and analysed. However, the manuscript lacks experimental support, and the simulated results have not been verified.
Author Response
This work demonstrates an application of ANSYS Composite Cure Simulation (ACCS) tool to simulate deformations and residual stresses induced in the composite structures due to the various design parameters used in manufacturing. Overall, the objectives of the paper are clear, and the methods and results are presented and analysed. However, the manuscript lacks experimental support, and the simulated results have not been verified.
Answer:
Thank you very much for your valuable comment and for reviewing our paper. We appreciate that you noted the lack of experimental results. Data has been added as you suggested. We have mentioned the experimental results provided by LMAT and Ameya [11, 45]. The results have been validated with experimental results and compared for the spring-in values observed due to the deformation in the composites. All the data has been elaborated in section 3.4 on line #437.

Reviewer 3 Report
The first paragraph of the introduction contains a few grammatical errors.
In the second paragraph of the introduction, claims have been made with no references.
References throughout the manuscript are not following a particular style. Authors are requested to unify them.
Line 121: the word material is not the correct jargon. It should either be referred to as a prepreg (before manufacturing) and as a composite (after manufacturing).
Line 122: What do you mean by the experimental simulation process?
Line 125: Could you elaborate more on what do you mean by Metallurgical properties?
Line 129: What do those dimensions represent? And why is the thickness higher?
On page 4, Table 1: the units are not mentioned clearly. (kg/m3) not kg m-3. The same goes for the other units too.
Furthermore, it would be easier to read the elastic and shear moduli if they are represented in GPa or MPa/
Why is there a fiber volume fraction in the resin properties? Is it the fiber volume fraction or resin volume fraction?
Line 146: the composite thickness was mentioned to be around 1.6mm whereas in line 129, it was about 5mm. Why are the values incoherent?
Line 155: what do you mean by fiber thickness? and why is it so high? For AS4 usually, the fiber diameters are about 8µm.
What will happen if the analysis is carried out in a stochastic way? In line 142, the authors mentioned that the temperature distribution is assumed to be homogenous but in reality, the local changes can also lead to shape distortions.
In line 158-159, authors mentioned that it is necessary to investigate the mesh convergence as it greatly influences the study. But in the manuscript, no such studies are presented. How was the mesh side selected? What was the element type and the nodes? Was it an adaptive meshing?
Figure 3 and 4 needs improvement, it is difficult to read the graphs with no X and Y labels and also no legend. Details would be helpful. Furthermore, what was the reason for selecting two different cure cycles? Usually, cooling rates also greatly influence the generation of residual stresses, why was that considered as a factor?
Line 212: Any reference to these equations?
Figure 8 should be improved. Proper axis labels and axis format.
Figure 9: Authors indicated that a difference in temperatures can be observed which is valid. However, in the figure, there is no great difference in outer and inner temperatures. A temperature difference of 0.044°C cannot be considered significant. Furthermore, the imported body temperature is about 30.6°C whereas your analysis is done from 20°C, is there a reason for this temperature difference?
In section 3.2, the first paragraph is repetitive and is not needed to be mentioned here.
Line 327: What was the actual shear strength of these composites? And the maximum shear stress was observed in which plane of reference (23 or31)? Depending on which plane you investigate the properties can vary. Moreover, 590MPa is a very high value. And what was the given layup?
https://link.springer.com/article/10.1007/s11340-010-9364-0
Authors should compare the achieved simulation results with existing experimental results to validate their approach.
Figure 11 part c: why is the geometry of the plate different?
Line 351: It was mentioned that the data was used to train the ANN. But there is no further information within the manuscript about this training or its predictions. What was the library used to set up the ANN model and what was the prediction accuracy? What was the input data? Nodal values or element averages or just the maximum and minimum values? In fact, the need for ANN is not clear. Since the authors carried out simulations for all the cases, why was there a need for ANN?
Table 3: What do authors mean by asymmetric layup? there is no information provided in the text. And in Line 155, the authors mentioned that in this study 2 different thicknesses are of interest but in the results section, no such information is available. Can the authors explain this discrepancy?
Why was the double cure cycle better than the single cure cycle? Was it because of the temperature homogeneity or resin induced effects? any references that can support this claim?
figure 14 is not a scatter plot, it is more of a line plot, figure 13 is a scatter plot.
Reference 3 and 25 are repeated in the manuscript.
Author Response
- The first paragraph of the introduction contains a few grammatical errors.
Answer:
With thanks, all errors are corrected.
- In the second paragraph of the introduction, claims have been made with no references.
Answer:
We thank the reviewer for a good feedback, some related references are added.
- References throughout the manuscript are not following a particular style. Authors are requested to unify them.
Answer:
With thanks, we fixed them.
- Line 121: the word material is not the correct jargon. It should either be referred to as a prepreg (before manufacturing) and as a composite (after manufacturing).
Thank you for pointing this out. The word is changed as suggested. Prepreg word used instead of material which is mentioned on line #124.
- Line 122: What do you mean by the experimental simulation process?
Answer:
Thanks for your comment, in the sentence on line # 124, we wanted to mention that the run (experiment) on Ansys Composite Cure Simulation (ACCS) is performed using the Hexcel material. Having said that, the sentence has been changed as ‘In this project, Hexcel AS4-8552 prepreg is used for the simulation process’. However, we experimented one case study to validate our results and discussed about it in sec. 3.4.
- Line 125: Could you elaborate more on what do you mean by Metallurgical properties?
Answer:
Thanks for pointing this out. We tried to explain here that Hexcel AS4-855 is better because of properties like high impact resistance, a reasonable translation of fiber properties, high strength, etc. We claimed this with Ameya’s study [11] research work. However, the metallurgical word does not fit here since we already mentioned all the properties separately in the sentence, and they are not all metallurgical, so we removed that word changed the sentence accordingly.
- Line 129: What do those dimensions represent? And why is the thickness higher?
Answer:
We appreciate your valuable comment. The L-plate dimensions are 86x50 x50 mm, represents the length, width, and flange height. For the given problem, the thickness used is 1.6 mm. Dimensions can be explained in the figure as shown below. The same figure is added in the manuscript as in figure 2(a). Dimensions written in the manuscript has been simplified as “L plate of dimensions 50x86x1.6 mm with flange height 50 mm”, which is mentioned on line #133
- On page 4, Table 1: the units are not mentioned clearly. (kg/m3) not kg m. The same goes for the other units too.
Answer:
Thanks. All units are changed as suggested.
- Furthermore, it would be easier to read the elastic and shear moduli if they are represented in GPa or MPa/
Answer:
Appreciate it. All units are changed, and values edited in GPa/Mpa.
- Why is there a fiber volume fraction in the resin properties? Is it the fiber volume fraction or resin volume fraction?
Answer:
Thank you for pointing out. It is a fiber volume fraction, and it should not be under the resin properties section. The data has been moved above the resin properties section.
- Line 146: the composite thickness was mentioned to be around 1.6mm whereas in line 129, it was about 5mm. Why are the values incoherent?
Answer:
Thanks for your meticulous review. There was confusion in the dimensions, but a new figure 2(a) and data mentioned on line #150 explains the thickness and other dimensions in detail. We already mentioned this change in response to comment 7.
- Line 155: what do you mean by fiber thickness? and why is it so high? For AS4 usually, the fiber diameters are about 8µm.
Answer:
Thanks for pointing that out. It is supposed to be fiber ply thickness. During the simulation, data are inserted in the ACP module of Ansys prepare the structure with the required composite properties.
- What will happen if the analysis is carried out in a stochastic way? In line 142, the authors mentioned that the temperature distribution is assumed to be homogenous but in reality, the local changes can also lead to shape distortions.
Answer:
Thanks for your good implication. Previous studies [11] in our lab discussed about some other parameters in the ACP package. They surveyed some literatures and assessment about the changes in the thermal processes and other prominent factors like Resin cure shrinkage, tool-part interaction that cause shape distortions. They showed that in most cases there would not be many differences in our output (shape deformations). However, we will look into this data in the next research (future work), where an artificial neural network will also get implemented in it.
- In line 158-159, authors mentioned that it is necessary to investigate the mesh convergence as it greatly influences the study. But in the manuscript, no such studies are presented. How was the mesh side selected? What was the element type and the nodes? Was it an adaptive meshing?
Answer:
Thank you very much for your note. We have added the detailed content about the mesh study in line # 167. Meshing with different element sizes and properties are used for the comparison (as shown in table2). The results show a significant difference in the accuracy of deformation values and simulation time. Considering optimized outputs, we selected a 1 mm element size for the current design with 8900 nodes and 8712 elements (details shown in the manuscript), which shows precision in the results with minimal simulation time. The shell elements can be used for the current structure. We have not selected an adaptive sizing option for the current work. The quality of the mesh is medium (smoothing).
Table 2. Significance of mesh size in the deformation
|
Element Size |
Nodes |
Elements |
Deformation |
Shear Stress |
|
0.5 |
35575 |
35199 |
1.14 |
96.676 |
|
1 |
8900 |
8712 |
4.6191 |
76.893 |
|
1.5 |
4080 |
3953 |
27.968 |
75.213 |
|
2 |
2295 |
2200 |
22.332 |
75.773 |
- Figure 3 and 4 needs improvement, it is difficult to read the graphs with no X and Y labels and also no legend. Details would be helpful. Furthermore, what was the reason for selecting two different cure cycles? Usually, cooling rates also greatly influence the generation of residual stresses why was that considered as a factor?
Answer:
Figure 3 and Figure 4 both improved by adding the X and Y axis labels with the units. Also, the details about the temperature change in the graph are provided on line #203.
We found several factors in the literature survey that can cause deformations and generate residual stresses in the composite structure. Few primaries of them were considered in this study. S. Hernandez [32] G.Fernlund [17] explains the effect of the cure cycle, layup in the autoclave process, whereas Ameya Patil [11] and White SR [31] mentioned the significance of the cure cycle in the shape deformations of composites. The double cure cycle showed better results than the single cure cycle and it is due to the resin induced effects. Literature on cure cycling implied that if the curing reaction of resin occurred too quickly the resin flow time will be reduced and resulted in voids, and deformations. It is essential to ensure complete cure of the final laminates, which guarantees a good laminate quality [19]. Dong [18] presented the study of lam-inate quality based on initial cure temperature and cure cycle used. They mentioned that the degree of cure increased with cure time until reaching a constant value. In the double hold cure cycle, the maximum degree of cure at 120 0C is 0.78 which suggests, an insufficient cross-linking network formation. When the temperature rises to 180 0C, degree of cure (DOC) reached 0.95, which results in a better laminate quality with less deformation. Also, the porosity in the laminate is minimum for the double hold cure cycle with dwell at 120 0C and 180 0C. While selecting the input parameters to find the optimal design, we considered the cure cycle as an important parameter based on previous studies.
- Line 212: Any reference to these equations?
Answer:
Reference has been added on line #221
- Figure 8 should be improved. Proper axis labels and axis format.
Answer:
With thanks, figure 8 replaced by the new one with improved axis labels units and quality.
- Figure 9: Authors indicated that a difference in temperatures can be observed which is valid. However, in the figure, there is no great difference in outer and inner temperatures. A temperature difference of 0.044°C cannot be considered significant. Furthermore, the imported body temperature is about 30.6°C whereas your analysis is done from 20°C, is there a reason for this temperature difference?
Answer:
We appreciate the reviewer’s constructive feedback. Figure 9 indicates the thermal analysis results by the Ansys ACCS tool. 20.044°C is the max temperature that may occur in the structure during the process.
The primary purpose to add this diagram and explanation is to show that the temperature is distributed from center to outer layer with max in the center.
The imported body temperature is shown in the static structural analysis after importing the load. The thermal loads obtained using the transient thermal analysis act as initial boundary conditions for the static structural analysis. When the load is imported, the ACCS tool shows the imported body temperature for the given design.
- In section 3.2, the first paragraph is repetitive and is not needed to be mentioned here.
Answer:
With thanks, paragraph shortened and changed as suggested.
- Line 327: What was the actual shear strength of these composites? And the maximum shear stress was observed in which plane of reference (23 or31)? Depending on which plane you investigate the properties can vary. Moreover, 590MPa is a very high value. And what was the given layup?
Answer:
We had calculated/considered a random layup to show the shear stress analysis in ACCS. The purpose of this graph is to explain the static structural analysis result shown in the ACCS tool. The 590MPa shear stress shown in the graph was for the [0/45/0] asymmetric layup. However, the new graph added with the deformations in the [0/45/90] symmetric layup. Figure 10 shows the maximum amount of shear stress and deformations that can be generated in the structure. The data has been added to line #329 to understand the graph clearly.
We have not considered the shear strength in the current study. Also, the Ansys ACCS tool shows the maximum shear stress value generated in the structure. For some designs, the stress value rises to 590 MPa. While for others, it shows less than 50 MPa.
- Authors should compare the achieved simulation results with existing experimental results to validate their approach.
Answer:
We have studied the previous data of case study and presented the comparison of simulation results with the experimental values. The information about the comparison and the detail values can be found in section 3.4 at line #437.
- Figure 11-part c: why is the geometry of the plate different?
Answer:
With thanks, the figure 11.C shows that the ACCS tool can be used for the complex composite parts to investigate the possible results. We tried to run the simulation on different designs. One of the complex structures (a gurney flap of a car) is as shown in the paper, and as mentioned on line 339, the design with a longer length may show the sagging and/or twist after curing. So, our case study is fig. 11 (a,b) and its dimension is fixed.
Line 351: It was mentioned that the data was used to train the ANN. But there is no further information within the manuscript about this training or its predictions. What was the library used to set up the ANN model and what was the prediction accuracy? What was the input data? Nodal values or element averages or just the maximum and minimum values? In fact, the need for ANN is not clear. Since the authors carried out simulations for all the cases, why was there a need for ANN?
Answer:
Thanks for the note. This paper primarily intends to the comparison of the ACCS results. Our current work is based on a strong foundation study on an innovative method developed to find the residual stresses. We have not trained the algorithm as yet. We would investigate the results in the future by using the Genetic Algorithm method (GA), in which we will also implement neural network (ANN).
We have mentioned an ANN in the paper to show the significance of ANN in the optimization and to show that we can develop an algorithm based on the results found in our study. However, line 351 is paraphrased for better understanding. We are grateful to the reviewer for the meticulous observation. In short, we wanted to know the feasibility of initiating a study associate with ANN. In the next phase of this study, we would use our inputs and outputs as building blocks for ANN.
- Table 3: What do authors mean by asymmetric layup? There is no information provided in the text. And in Line 155, the authors mentioned that in this study 2 different thicknesses are of interest but in the results section, no such information is available. Can the authors explain this discrepancy?
Answer:
It is noted in the experiment that the sequence of fiber orientations leads to a change in the deformation values. The asymmetric layup can be defined as the layup of which the first three layups of the given design are not similar/ in sequence with the next three. For example, [0/45/90]as can be defined as [0/45/90/90/-45/0].
In the current project work, the fiber thickness kept constant as 1.6mm since the main focus of this research work was to find the optimal design with the minimum deformation. The objective was to compare the same size design based on different processes and parameters included in the manufacturing. A change in the layup thickness could not give the parameter (base) to compare the design.
- Why was the double cure cycle better than the single cure cycle? Was it because of the temperature homogeneity or resin induced effects? any
Answer:
It is because of the resin induced effects. We have studied the cure cycle effect through the literature survey. If the curing reaction of resin occurred too quickly the resin flow time will be reduced and resulted in voids, and deformations. It is essential to ensure complete cure of the final laminates, which guarantees a good laminate quality [19]. Dong [18] presented the study of laminate quality based on initial cure temperature and cure cycle used. They mentioned that the degree of cure increased with cure time until reaching a constant value. In the double hold cure cycle, the maximum degree of cure at 120 0C is 0.78 which suggests, an insufficient cross-linking network formation. When the temperature rises to 180 0C, DOC reached 0.95, which results in a better laminate quality with less deformation. Also, the porosity in the laminate is minimum for the double hold cure cycle with dwell at 120 0C and 180 0C.
- figure 14 is not a scatter plot, it is more of a line plot, figure 13 is a scatter plot.
Answer:
Thank you for the note. Figure 14 is changed with clustered column graph.
- Reference 3 and 25 are repeated in the manuscript.
Answer:
With thanks, it has been fixed now.

Round 2
Reviewer 1 Report
The paper is entitled "Optimization of Composite Structures in Curing Process for Minimum Shape Deformations and Shear Stress". The topic of high practical relevance and clearly within the scope of the journal.
Unfortunately, the reviewer still sees greater shortcomings in the revised paper. So, bevor considering publication another mayor revision is necessary.
Some remarks more in detail:
In their response to the first review of the reviewer the authors state "We have not trained the algorithm as yet. We would investigate the results by using the Genetic Algorithm (GA) in the future, in which we will also implement an artificial neural network (ANN). We have added some contents about the ANN to show its significance in the optimization process and to show that we can develop an algorithm based on the results found in our study. (…) Instead, we focused on the comparison and stated that the foundation of this study and input and output parameters are ready to get implemented in ANN to find the optimal structure. In another word, the data executed in the FEA tool would be utilized to train the parametric data in artificial neural network." In this case, the title is misleading and raises expectations that are not fulfilled. Therefore, the reviewer recommends, to change the title in such a way that it is clarified, that here the basis for the application of optimization methods (like GA using ANN) .
The same is true for the abstract. A sentence like "This paper presents an investigation of process-induced shape deformations in the composite parts and structures and implementation of the global optimization techniques to finalize the design parameters." raises the same false expectations. Therefore, the abstract has to become clearer.
Line 84, paragraph 76 - 96:
"This paper shows the study of the design optimization technique for the composite parts by considering the manufacturing limitations." In the opinion of the reviewer this study shows a basis (or fundament) for the application of design optimization not a design optimization technique. So the paragraph 76 - 96 should be reworked.
Figure 1:
Please check, if the arrow between "Execute FEA" and "Evaluate objective function" is correct.
Table 2:
Depending on the element size, the deformation and shear stress are "wildly" commuting forth and back. Please comment on this and explain more clearly, why an average element size of 1 mm was chosen. Looking on the numbers, the reviewer would doubt, that the FE-model is really converging against the correct result.
Line 225:
"This research work presents a composite structure's optimization by considering limitations in the manufacturing processes and material." Please correct: "This research work presents the basis for a composite structure's optimization"
General remark:
The authors stat in their response to the first review, that they did not use an optimization method on their results but the results can be used as input for an optimization, as to the understanding of the reviewer. This should be clear in the whole article. So please check the whole text carefully for misleading expressions!
Line 251:
As the ANN has not been implemented and trained, it is not clear, that a "MLP network with three hidden layers" is suitable for the problem. In addition, the ANN does not have two output layers, but one output layer (as shown in figure 7) with 2 neurons.
Line 327:
Please point out, the here the results for the given layup are presented exemplarily
Line 350:
In the opinion of the reviewer, as no optimization but a parameter study, based on the parameter values being chosen by LHS, is carried out teh section should be titled "Parameter study results"
Figure 12:
Is not showing an NN-diagram but a diagram of parameter combinations studied in the scope of this work!
Line 371, Figure 13, Table 5:
The nomenclature "Design_xx" is not explained. No relation between the parameter combinations given in table 4 and the descriptions Design_??" given in table 5 are obvious.
Table 6:
Please identify upper row and lower row within the combining rows "Deformation" and "Shear Stress"
Figure 15:
On the x-axis the different layup-configurations are referenced. These are only discrete points and the values are not continuously distributed. Therefore in the opinion of the reviewer it is not possible to generate a graph by connecting the y-values with a line.
(see Remarks on Figure 14 in first review)
Figure 16:
See figure 15
Finally, A carful correction reading is recommended, for there seem to be some tipping errors (upper vs. lower letters, etc).
Author Response
The paper is entitled "Optimization of Composite Structures in Curing Process for Minimum Shape Deformations and Shear Stress". The topic of high practical relevance and clearly within the scope of the journal.
Unfortunately, the reviewer still sees greater shortcomings in the revised paper. So, bevor considering publication another mayor revision is necessary.
Some remarks more in detail:
- In their response to the first review of the reviewer the authors state" We have not trained the algorithm as yet. We would investigate the results by using the Genetic Algorithm (GA) in the future, in which we will also implement an artificial neural network (ANN). We have added some contents about the ANN to show its significance in the optimization process and to show that we can develop an algorithm based on the results found in our study. (…) Instead, we focused on the comparison and stated that the foundation of this study and input and output parameters are ready to get implemented in ANN to find the optimal structure. In another word, the data executed in the FEA tool would be utilized to train the parametric data in artificial neural network." In this case, the title is misleading and raises expectations that are not fulfilled. Therefore, the reviewer recommends, to change the title in such a way that it is clarified, that here the basis for the application of optimization methods (like GA using ANN).
The same is true for the abstract. A sentence like "This paper presents an investigation of process-induced shape deformations in the composite parts and structures and implementation of the global optimization techniques to finalize the design parameters." raises the same false expectations. Therefore, the abstract has to become clearer.
Answer-
We completely agree with the reviewer’s opinion, the title has been changed to “Analysis of Composite Structures in Curing Process for Shape Deformations and Shear Stress: Basis for Advanced Optimization” Abstract also edited with the suggested changes.
- Line 84, paragraph 76 - 96:
"This paper shows the study of the design optimization technique for the composite parts by considering the manufacturing limitations." In the opinion of the reviewer this study shows a basis (or fundament) for the application of design optimization not a design optimization technique. So the paragraph 76 - 96 should be reworked.
Answer-
The paragraph is changed with the suitable information. The new content can be found in line# 76 to 104
- Figure 1:
Please check, if the arrow between "Execute FEA" and "Evaluate objective function" is correct.
Answer-
Thanks for making it clear and easier. The arrow between Execute FEA" and "Evaluate objective function” has been removed.
- Table 2:
Depending on the element size, the deformation and shear stress are "wildly" commuting forth and back. Please comment on this and explain more clearly, why an average element size of 1 mm was chosen. Looking on the numbers, the reviewer would doubt, that the FE-model is really converging against the correct result.
Answer-
Thank you very much for this precious comment and for showing your concern about the element size. Here, we try to elaborate why a 1 mm element size is being selected. We searched for a balance between computing time and accuracy.
Selecting a bigger element size (1.5 or 2mm) may have shown a convergence, but the results that we have found for the deformation is not compliant with similar studies [11,46,47]. For example, the 1.5 mm element size shows 27.968 mm deformation which in turn shows bigger spring-in results that are not compliant with the actual results (more discussed in section 3.4). So, technically bigger element size would not function properly to show the actual deformation results which is the primary objective of our research work.
Addressing about refined mesh/element size (0.5 mm), usually smaller mesh means more accurate results, but the computing time gets significant as well. We also noticed through the simulation that the shear stress increases drastically as we go with finer element size. Unfortunately, in almost all analysis performed for commercial or scientific purposes the solution of the problem is unknown. In such cases the “typical” approach doesn’t work. Instead, we had to “guess” the correct answer basing on the models with different meshes we have done. When we reduced it to lower than 1 mm, most of simulations went erupted. So, we could not trust to data of .5 mm.
Based on the shear stress and deformation results, 1 mm element size gives the satisfactory values (eg. 4.61mm) which can be analyzed and used to compensate during the part manufacturing. Finally, it should be noted this Ansys Package (ACCS) is very new in market and there are no studies and information on how singularities can be dealt with
- Line 225:
"This research work presents a composite structure's optimization by considering limitations in the manufacturing processes and material." Please correct: "This research work presents the basis for a composite structure's optimization"
Answer-
We thank the editor for the comment. Corrected information can be found in line# 230
- General remark:
The authors state in their response to the first review, that they did not use an optimization method on their results but the results can be used as input for an optimization, as to the understanding of the reviewer. This should be clear in the whole article. So please check the whole text carefully for misleading expressions!
Answer-
Thank you for the remark. We carefully checked the content, and the changed where it’s necessary.
- Line 251:
As the ANN has not been implemented and trained, it is not clear, that a "MLP network with three hidden layers" is suitable for the problem. In addition, the ANN does not have two output layers, but one output layer (as shown in figure 7) with 2 neurons.
Answer-
We are grateful for this crucial remark. We framed the new sentence which is suitable for the topic. The sentence is “A Multi-Layers Perceptron (MLP) network with three hidden layers and one output layer having two neurons, CAN BE defined for the given problem”. What we want to explain here is that when we will develop ANN for the given problem, we may be able to generate the network with 3 hidden layers (Curing Cycle, Symmetry, and Constraints). Also, about the last part, the line is changed as following – “Two neurons are the objectives with the minimum deformation and shear stress.”
Line 327:
Please point out, the here the results for the given layup are presented exemplarily.
Answer-
Thanks for your good tip. The sentence is added and highlighted with red in section 3.1.
- Line 350:
In the opinion of the reviewer, as no optimization but a parameter study, based on the parameter values being chosen by LHS, is carried out the section should be titled "Parameter study results"
Answer-
Thanks for suggesting the important change. The title has been changed as suggested. (section 3.3)
- Figure 12:
Is not showing an ANN-diagram but a diagram of parameter combinations studied in the scope of this work!
Answer-
Thanks for the observing a small but important error. The caption has changed as “The details of categories that can be considered to build the neural network”.
- Line 371, Figure 13, Table 5:
The nomenclature "Design_xx" is not explained. No relation between the parameter combinations given in table 4 and the descriptions Design_??" given in table 5 are obvious.
Answer-
With thanks, Table 4 edited with the data and Design nomenclature has added in the Sr No so that it can be concise with table5.
- Table 6:
Please identify upper row and lower row within the combining rows "Deformation" and "Shear Stress"
Answer-
Table 6 is replaced with a new layout with readable and convenient format.
- Figure 15:
On the x-axis the different layup-configurations are referenced. These are only discrete points and the values are not continuously distributed. Therefore in the opinion of the reviewer it is not possible to generate a graph by connecting the y-values with a line.
(see Remarks on Figure 14 in first review)
Figure 16:
See figure 15
Answer-
Thank you recognizing this change, we missed it in the last review. Figure 15 has been replaced with column type of graph. Figure 16 is changed similar to the figure 14 and figure 15.
- Finally, A carful correction reading is recommended, for there seem to be some tipping errors (upper vs. lower letters, etc).
Answer-
We tried our best to correct the tipping errors in the manuscript.

Reviewer 2 Report
The authors have improved the paper according to the comments. I have no further comments.
Author Response
Thanks for your approval.
Reviewer 3 Report
Thank you for answering my long list of comments.
However, I still have one question.
Point 18 in Reviewer #3 comments: You have answered that 20.004°C is the maximum temperature that the structure experiences during the process. Why is it only 20°C when the curing cycle is at 180°C?
I think it is the temperature after demolding i.e., at the end of the processes? That correction should be made in that statement.
Author Response
Comments and Suggestions for Authors
Thank you for answering my long list of comments. However, I still have one question.
Point 18 in Reviewer #3 comments: You have answered that 20.004°C is the maximum temperature that the structure experiences during the process. Why is it only 20°C when the curing cycle is at 180°C?
I think it is the temperature after demolding i.e., at the end of the processes. That correction should be made in that statement.
Answer-
Thanks for the review and valuable comment. It is a temperature of the structure at the final stage in the thermal analysis. In the convection process, according to cure cycle, the temperature raises from 20 to 180 and hold for certain time (depends on the cycle) and then cools down to 20 again. However, in the last stage temperature of the given structure can rise upto 20.044 0C. We have corrected the data in the manuscript. The updated content can be found in line#322

Round 3
Reviewer 1 Report
The paper is entitled "Analysis of Composite Structures in Curing Process for Shape Deformations and Shear Stress: Basis for Advanced Optimization". The topic of high practical relevance and clearly within the scope of the journal.
After the changes in the latest revision, I recommend the acceptance of the paper in the present form.